

# Imbalance in the blood antioxidant system in growth hormone-deficient children before and after 1 year of recombinant growth hormone therapy

Maria S. Pankratova[1], Adil A. Baizhumanov[2], Alexander I. Yusipovich[2], Maria Faassen[1], Tatyana Yu. Shiryaeva[1], Valentina A. Peterkova[1], Svetlana S. Kovalenko[2], Tatiana A. Kazakova[2] and Georgy V. Maksimov[2]

[1] Department of Paediatric Endocrinology, Endocrinology Research Centre, Moscow, Russian Federation
[2] Faculty of Biology, Department of Biophysics, Lomonosov Moscow State University, Moscow, Russian Federation

Corresponding author
Alexander I. Yusipovich,
sanyavor@gmail.com

## ABSTRACT

The aim of our study was to examine the effects of 12-month therapy with recombinant growth hormone (rGH) on the blood antioxidant system in children with growth hormone deficiency (GHD). Total antioxidant capacity (TAC) of plasma was measured by FRAP (ferric reducing antioxidant power or ferric reducing ability of plasma); activities of superoxide dismutase (SOD) and catalase (CAT) in erythrocytes were assessed; non-protein thiols (NT) and ceruloplasmin (CP) levels were also measured. These parameters were determined before and after 12 month of rGH treatment. Eleven treatment-naive prepubertal children with growth hormone deficiency were included in the study. Another 11 prepubertal children comprised a control group. Before rGH treatment, TAC of plasma and NT level in the control group were significantly lower ($726 \pm 196$ vs. $525 \pm 166$ μmol/L, $P = 0.0182$ and $0.92 \pm 0.18$ vs. $0.70 \pm 0.22$ μmol/ml, $P = 0.0319$, before and after the therapy, respectively). The only parameter that significantly ($19.6 \pm 4.7$ vs. $14.5 \pm 3.4$ Units/g Hb, $P = 0.0396$) exceeded the same in the control group after rGH therapy was SOD activity. However, none of the measured parameters of antioxidant system in GHD children, except for TAC ($525 \pm 166$ vs. $658 \pm 115$ μmol/L, $P = 0.0205$), exhibited significant improvement toward the end of the 12-month treatment period, although non-significant changes in CAT activity and CP level were also observed. This work has demonstrated that some parameters of the blood antioxidant system are out of balance and even impaired in GHD children. A 12-month treatment with rGH resulted in a partial improvement of the antioxidant system.

## INTRODUCTION

Free radicals and other reactive species are thought to play an important role in many human diseases. A serious imbalance between production of reactive species and the antioxidant protective system due to increased production of reactive species or low levels of antioxidants leads to oxidative damage (oxidative stress, OS) and development of various disorders (*Halliwell & Whiteman, 2004*). Therefore, the evaluation of OS could be used as a nonspecific marker of systemic disorders in the human body. Moreover, if the standard treatment shows itself as not sufficient to decrease an ongoing OS, the applied therapy may require modification, for example, by additional administration of vitamins, antioxidants, etc. We believe that some parameters of the blood antioxidant status could be used for evaluation of OS.

It is particularly important to estimate OS in GHD children. Treatment with recombinant growth hormone (rGH) and a subsequent increase in linear growth rate can lead to acceleration of metabolic processes and be accompanied by changes in the blood antioxidant status parameters. Furthermore, an imbalance of the antioxidant parameters was reported in some studies (*Evans et al., 2000*; *Gonzalez-Duarte et al., 2012*), where patients with adult GHD demonstrated a high degree of OS. Unfortunately, as of today only a small number of publications are dedicated to OS in GHD children, particularly those subjected to treatment with rGH. In fact, we are aware of only one paper (*Mohn et al., 2005*). This paper demonstrated that OS parameters (index of susceptibility of low-density lipoprotein (LDL) to *in vitro* oxidation, malondialdehyde and vitamin E levels) in GHD children were substantially higher than those in the healthy control group, whereas after one year of rGH therapy these parameters returned to normal levels.

Therefore, in this study we evaluated several parameters of the blood antioxidant system: total antioxidant capacity (TAC) of plasma, activities of superoxide dismutase (SOD) and catalase, non-protein thiols (NT) and ceruloplasmin (CP) levels. These parameters can help to reveal OS in children with GHD and examine the effects of 12-month rGH therapy on the blood antioxidant system.

## MATERIALS AND METHODS

Eleven treatment-naive GH-deficient patients were included in this study (2 girls and 9 boys aged 3–9 years). The mean chronological age (CA) was $6.1 \pm 2.2$ years, mean bone age (BA) was $2.6 \pm 0.9$ years. None of them has ever undergone treatment with rGH.

The parameters of the blood antioxidant system were compared with those in a control group of 11 healthy prepubertal children (2 girls and 9 boys; aged 6–11 years; mean CA $9.3 \pm 1.4$ years). The control group did not receive any placebo injections.

Standard deviation scores (SDS) for height and growth velocity were calculated using mean and standard deviation of the British reference population as described by *Tanner, Whitehouse & Takaishi (1966)* and *Tanner & Whitehouse (1976)*. Pubertal stages are defined accordingly to *Tanner (1962)*.

The SDS values for IGF-1 and IGFBP-3 were calculated for corresponding CA and gender (*Lofqvist et al., 2001*; *Lofqvist et al., 2004*).

All patients underwent a standard set of clinical and laboratory tests including physical and anthropometric evaluations, x-rays of both hands and wrists, as well as anCT or MRI of the head. To verify the diagnosis, GH-provocation tests were performed: 5 samples were evaluated with clonidine (0, 30, 60, 90, and 120 min), and 7 samples were evaluated with insulin (0, 15, 30, 45, 60, 90, and 120 min). A value of less than 10 ng/ml was an argument in support of the GHD diagnosis (*GH Research Society , 2000*). Peak values of GH response in both tests were $1.71 \pm 1.45$ ng/ml (with the minimum and maximum values 0.13 and 3.5 accordingly). Idiopathic isolated GH-deficiency was diagnosed in all the cases studied. Clinical and biochemical blood analyses, as well as assays for IGF-1 and IGF binding protein-3 (IGFBP-3) were performed before treatment and after 12 months of rGH treatment. The IGF-1 levels were measured by immunoradiometric assay using a commercial IGF-1 RIA (Nichols Institute Diagnostics, Bad Nauheim, Germany); IGFBP-3 level was determined by enzyme immunoassay using commercial DSL-10-6600 ACTIVE$^{TM}$ IGFBP3 ELISA kit (DSL, Sinsheim, Germany). Daily rGH injections were subcutaneously administered in the evening. The daily dose of rGH was 0.033 mg per kg body weight (*GH Research Society , 2000*).

Analyses of the antioxidant system were performed on whole blood samples, which were collected in the fasting state in the morning, before and after 12 months of rGH treatment.

## Antioxidant status evaluation

The erythrocyte SOD activity was estimated by inhibition of epinephrine self-oxidation at 25 °C (*Sun & Zigman, 1978*). One unit of SOD activity was defined as that amount of SOD required to cause 50% inhibition of the oxidation of the epinephrine. The SOD activity was expressed as units per gram of haemoglobin (Units/g Hb).

The level of ceruloplasmin was estimated by measuring the enzymatic reaction with o-phenylenediamine (*Brazhe et al., 2014*).

Catalase activity was measured in erythrocytes at 37 °C according to *Aebi (1984)*. One unit of catalase activity (k) was defined as the amount of enzyme that degrades one mmol $H_2O_2$/min at initial concentration of 10 mmol/L at pH 7.0. The catalase activity was expressed as the unit per gram hemoglobin (k/g Hb).

Non-protein thiols were analyzed as described by *Sedlak & Lindsay (1968)* with modifications (*Akhalaya, Platonov & Baizhumanov, 2006*).

The TAC in the blood plasma was estimated by the ferric reducing ability of plasma (FRAP) assay, as described by *Benzie & Strain (1996)* with modifications: 350 µL of distilled water was added to the test tube containing 3 mL of the reagent (working solution), then 50 µL of plasma sample was added and mixed. After 10 min the samples were read at a wavelength of 593 nm. The method is based on the reduction of colorless ferric ($Fe^{3+}$) tripyridyltriazine complex in working solution to blue colored ferrous ($Fe^{2+}$) tripyridyltriazine complex at low pH. The TAC values were obtained by comparing the absorption change in the test mixture with those obtained from increasing concentrations of $Fe^{2+}$ and expressed as µmol of $Fe^{2+}$ equivalents per L of sample.

Photometric assay of the blood hemoglobin is based on the transformation of hemoglobin into its haemachrome form by sodium dodecyl sulphate, followed by absorption of the measuring light at 540 nm (*Brazhe et al., 2014*).

Changes in the optical density were recorded with a Hitachi-556 spectrophotometer (Hitachi, Tokyo, Japan).

## Statistics

The results were statistically processed using Statistica software, version 8.0. All data were normally distributed (Kolmogorov–Smirnov test, $p < 0.05$). Statistical significance of differences for independent variables, $P_1$ (between parameters of control group and parameters of case group before treatment) and $P_2$ (between parameters of control group and parameters of case group after treatment), were calculated by one-way ANOVA with the post-hoc Tukey HSD test. The statistical significance of differences for dependent variables, $P_3$ (between parameters of case group before and after rGH treatment) only for case group, was calculated by the paired $T$-test. Changes were considered significant at $P < 0.05$.

The power (one-tailed dependent and one-tailed independent $T$-test at $\alpha = 0.05$ and the sample size of 11) was calculated using the Power Analysis box of Statistica 8.0.

## Ethics statement

The study was approved by the Ethics Committee of the Endocrinology Research Centre, Moscow, Russian Federation (reference number: 14). Written informed consent was obtained from the patients and/or their parents or legal guardians.

## RESULTS

### Anthropometric and biochemical parameters

Children treated with rGH demonstrated increases in height, weight, height velocity, height SDS, and height velocity SDS (Table 1). To assess therapeutic efficacy, safety and compliance, IGF-1 SDS and IGFBP-3 SDS were assessed, which showed clear increases during rGH treatment.

Growth parameters after 12 months of rGH therapy significantly exceeded those at baseline (Table 1). After 12 months of treatment the patients' height was significantly increased ($96.2 \pm 10.5$ vs. $108.0 \pm 10.8$ cm, $P_3 = 0.000001$), and the increase in growth velocity ($3.4 \pm 1.2$ vs. $12.5 \pm 3.5$ cm/years, $P_3 = 0.000013$) suggested that catch-up growth was significant. Thus, catch-up in growth was obvious: height SDS increased from $-3.6 \pm 0.9$ at the onset of therapy and to $-2.2 \pm 1.3$ after 12 months ($P_3 = 0.000030$). Serum IGF-1 SDS level was $-3.1 \pm 1.8$ at baseline and increased to $-0.7 \pm 1.9$ after 12 months of treatment ($P_3 = 0.000689$); IGFBP-3 SDS increased from $-4.0 \pm 3.5$ to $-0.2 \pm 1.7$ after treatment ($P_3 = 0.002364$).

### The antioxidant status

To evaluate the blood antioxidant status we selected the most characteristic parameters: total antioxidant capacity (TAC) of plasma measured by FRAP, superoxide dismutase (SOD) and catalase activities, non-protein thiols (NT) and ceruloplasmin levels. The FRAP

**Table 1** Effect of growth hormone therapy on anthropometric and biochemical parameters of GHD children.

| Parameters | Before GH therapy | After 12 months of GH therapy |
| --- | --- | --- |
| Height, cm | $96.2 \pm 10.5$ | $108.0 \pm 10.8$<br>$P_3 = 0.000001^*$ |
| Weight, kg | $15.2 \pm 3.9$ | $17.8 \pm 5.5$<br>$P_3 = 0.001118^*$ |
| Growth velocity, cm/years | $3.4 \pm 1.2$ | $12.5 \pm 3.5$<br>$P_3 = 0.000013^*$ |
| Height SDS | $-3.6 \pm 0.9$ | $-2.2 \pm 1.3$<br>$P_3 = 0.000030^*$ |
| Growth velocity SDS | $-3.2 \pm 1.8$ | $7.4 \pm 3.9$<br>$P_3 = 0.000026^*$ |
| IGF-1 SDS | $-3.1 \pm 1.8$ | $-0.7 \pm 1.9$<br>$P_3 = 0.000689^*$ |
| IGFBP-3 SDS | $-4.0 \pm 3.5$ | $-0.2 \pm 1.7$<br>$P_3 = 0.002364^*$ |

**Notes.**

$P_3$ indicates the statistical significance between parameters of the case group before and after rGH treatment, using the paired $T$-test, $P < 0.05$

[*] Significant difference between data.

is proportional to the reducing power of the main non-enzymatic antioxidants in the plasma, particularly uric and ascorbic acids. This parameter does not reflect the reduced glutathione and liposoluble antioxidants (e.g., vitamin E). Ceruloplasmin and SOD are responsible for both utilization of the superoxide anion radical and the regulation of variable-valence metal levels (copper and iron). Catalase and non-protein thiols play an important role in hydrogen peroxide utilization.

The parameters of blood antioxidant system (as mean $\pm$ standard deviation) for control and case (before and after rGH treatment) groups are shown in Table 2. Before treatment, the TAC of plasma and the amount of NT were significantly lower than in the control group (Fig. 1): $525 \pm 166$ vs. $726 \pm 196$ μmol/L and $0.70 \pm 0.22$ vs. $0.92 \pm 0.18$ μmol/ml ($P_1 = 0.0182$ and $P_1 = 0.0319$), respectively. After rGH therapy only SOD activity differed from that in the control group ($19.6 \pm 4.7$ vs. $14.5 \pm 3.4$ Units/g Hb, $P_2 = 0.0396$).

The paired $T$-test revealed no significant improvements in most of the parameters except for the TAC of plasma at the end of the 12-month treatment period (see Fig. 2). The TAC value increased significantly (by 30%) during treatment (from $525 \pm 166$ to $658 \pm 115$, $P_3 = 0.0205$). The increase in NT level and SOD activity was not statistically significant ($0.70 \pm 0.22$ vs. $0.78 \pm 0.15$ μmol/ml, $P_3 = 0.3264$ and $18.8 \pm 5.4$ vs. $19.6 \pm 4.7$ Units/g Hb, $P_3 = 0.7436$, respectively), and the decrease in catalase activity and ceruloplasmin level during treatment were also not statistically significant ($215 \pm 64$ vs. $190 \pm 43$ k/g Hb $P_3 = 0.2257$ and $581 \pm 100$ vs. $531 \pm 111$ μg/mL $P_3 = 0.1061$, respectively).

## Statistical power

The statistical power for all measured parameters was calculated (Table 3). The power varied greatly for different parameters. The highest power was revealed for anthropometric

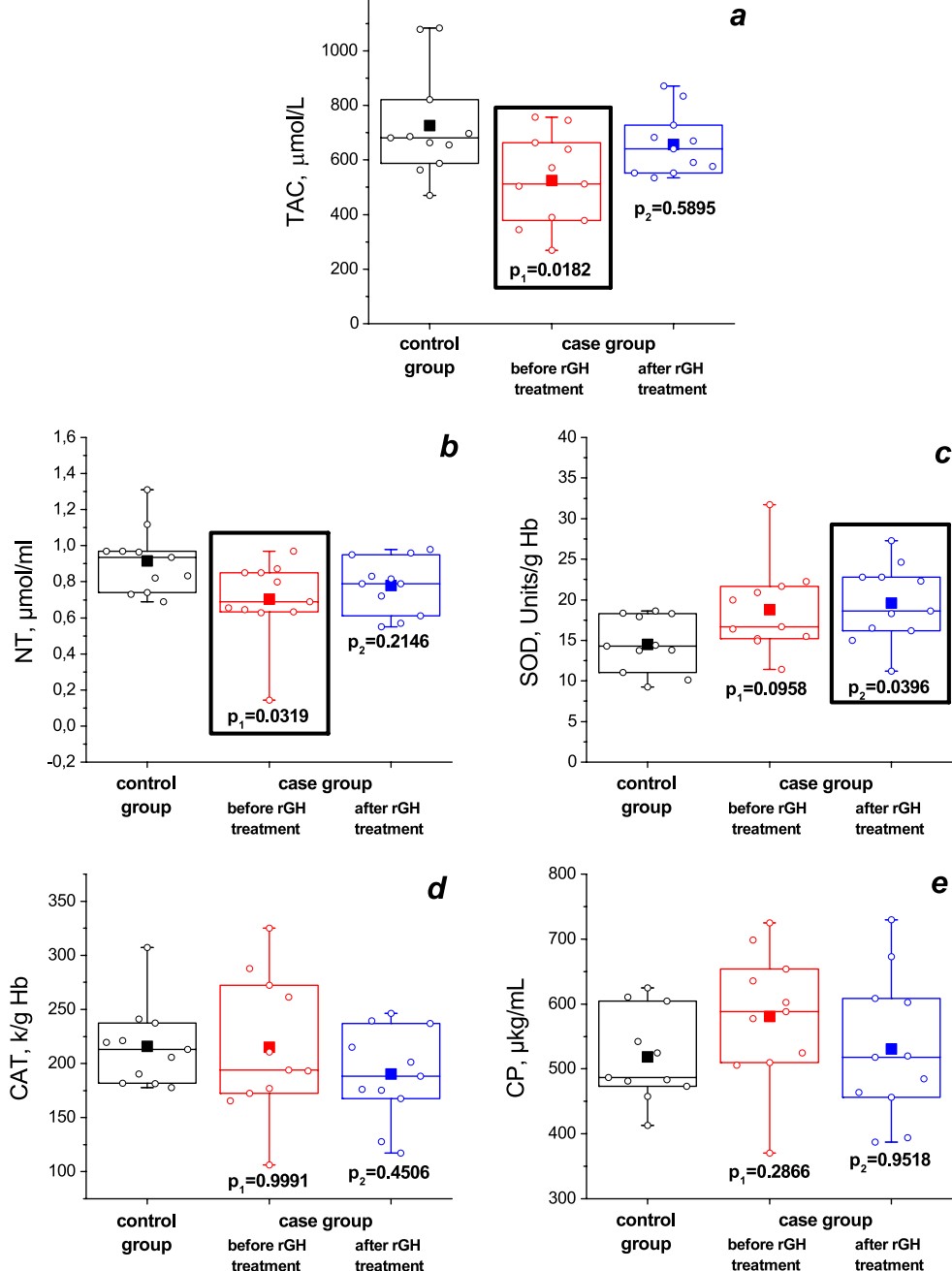

**Figure 1 Changes in TAC (A), NT level (B), erythrocyte SOD (C) and catalase (D) activities, CP level (E) after a 12 month rGH treatment (case-control data).** The data presented as a standard box and whisker plot (where the length of the box is an interquartile range) with the mean (closed square) and experimental data (open circle), the ends of the whiskers represent the maximal and minimal values. In the picture data of the control group (black box) and that of the case group before (red box) and after (blue box) rGH treatment is shown. $P_1$ is the statistical significance between parameters of the control group and those of the case group before treatment; $P_2$ is the statistical significance between parameters of the control group and those of the of case group after treatment. The experimental data were evaluated by the one-way ANOVA with post-hoc Tukey HSD test. The significant difference ($P < 0.05$) for $P_1$ and $P_2$ is shown by black rectangles.

**Table 2** Effect of growth hormone therapy on parameters of the blood antioxidant status of GHD children.

| Parameters | Control | Before GH therapy | After 12 months of GH therapy |
|---|---|---|---|
| Total antioxidant capacity of plasma, μmol/L | $726 \pm 196$ | $525 \pm 166$<br>$P_1 = 0.0182^*$ | $658 \pm 115$<br>$P_2 = 0.5895$<br>$P_3 = 0.0205^*$ |
| Non-protein thiols, μmol/ml | $0.92 \pm 0.18$ | $0.70 \pm 0.22$<br>$P_1 = 0.0319^*$ | $0.78 \pm 0.15$<br>$P_2 = 0.2146$<br>$P_3 = 0.3264$ |
| Superoxide dismutase (SOD), Units/g Hb | $14.5 \pm 3.4$ | $18.8 \pm 5.4$<br>$P_1 = 0.0958$ | $19.6 \pm 4.7$<br>$P_2 = 0.0396^*$<br>$P_3 = 0.7436$ |
| Catalase, k/g Hb | $216 \pm 38$ | $215 \pm 64$<br>$P_1 = 0.9991$ | $190 \pm 43$<br>$P_2 = 0.4506$<br>$P_3 = 0.2257$ |
| Ceruloplasmin, μg/mL | $518 \pm 70$ | $581 \pm 100$<br>$P_1 = 0.2866$ | $531 \pm 111$<br>$P_2 = 0.9518$<br>$P_3 = 0.1061$ |

Notes.

$P_1$ indicates the statistical significance between parameters of the control group and those of the case group before treatment, $P_2$ is the statistical significance between parameters of the control group and those of the case group after treatment; values were evaluated using the one-way ANOVA with post-hoc Tukey HSD test, $p < 0.05$. $P_3$ is the statistical significance between parameters of case group before and after rGH treatment, using the paired $T$-test, $p < 0.05$.

* Significant differences between data.

**Table 3** The calculated values of statistic power.

| Parameter | Power |
|---|---|
| **Dependent data (one-tailed dependent $T$-test α = 0.05, the sample size-11)** | |
| Height, cm | 1.00 |
| Weight, kg | 0.99 |
| Growth velocity | 1.00 |
| Height SDS | 1.00 |
| Growth velocity SDS | 1.00 |
| IGF-1, nMol/L | 0.92 |
| IGFBP-3, nMol/L | 1.00 |
| IGF-1 SDS | 1.00 |
| IGFBP-3 SDS | 0.98 |
| Total antioxidant capacity of plasma, μmol/L | 0.82 |
| Non-protein thiols, μmol/ml | 0.25 |
| Superoxide dismutase (SOD), Units/g Hb | 0.09 |
| Catalase, k/g Hb | 0.33 |
| Ceruloplasmin, μg/mL | 0.50 |
| **Independent data (one-way ANOVA α = 0.05, the sample size—11)** | |
| Total antioxidant capacity of plasma, μmol/L | 0.62 |
| Non-protein thiols, μmol/ml | 0.56 |
| Superoxide dismutase (SOD), Units/g Hb | 0.57 |
| Catalase, k/g Hb | 0.20 |
| Ceruloplasmin, μg/mL | 0.26 |

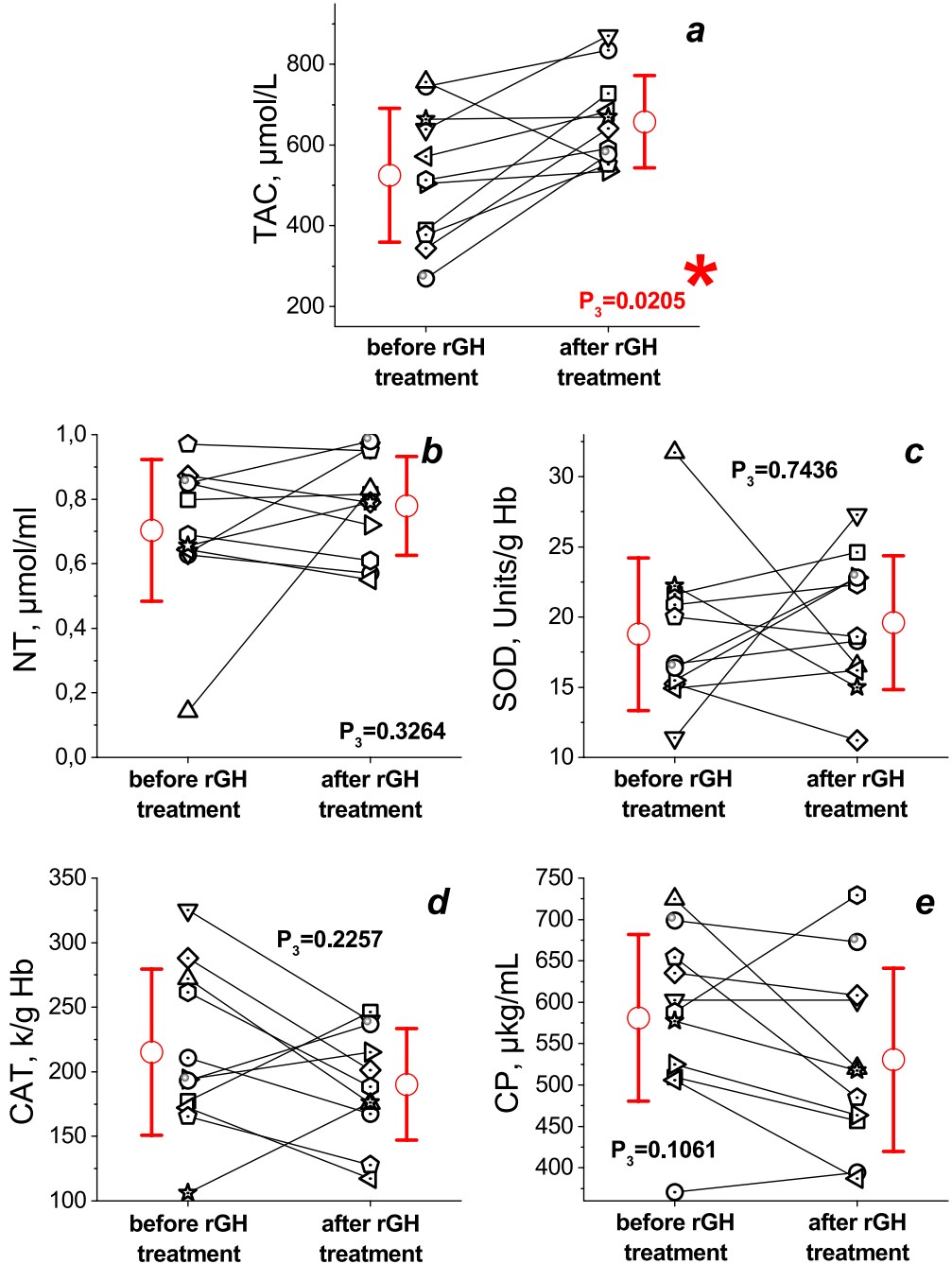

**Figure 2 Changes in TAC (A), NT level (B), erythrocyte SOD (C) and catalase (D) activities, CP level (E) during the 12 month rGH treatment (longitudinal data).** The data of case group before and after rGH treatment is shown in picture. Different symbols corresponds to the parameters obtained in various patients before and after the treatment (data of each patient is connected by line). The mean values ± standard deviations for case group before and after treatment is shown as open red circles with whiskers. $P_3$ is the statistical significance between parameters of case group before and after rGH treatment. The experimental parameters were evaluated by the paired $T$-test, $p < 0.05$. Significant difference between data is marked by an asterisk.

and biochemical parameters (height, weight, growth velocity, height SDS, growth velocity SDS, IGF-1, and IGFBP-3), which exceeded 0.91. For parameters of the blood antioxidant system, statistical power was considerably smaller. In this case, the most profound changes were found for TAC during GH treatment (0.82), and also in comparison of TAC, NT and SOD between the control and experimental groups (0.62, 0.56 and 0.57, respectively). The lowermost powers were found in changes in SOD and NT in the experimental group during rGH therapy (0.09 and 0.25, respectively) and also for the comparison of catalase and CP between the control and treatment groups (0.20 and 0.26, respectively).

## DISCUSSION

High power was found for anthropometric and biochemical parameters, in contrast to parameters of the antioxidant system. In general, we revealed small differences in the mean values of antioxidant system parameters and relatively high dispersion (standard deviation), which gives a rather high type 2 error for the small sample volume of 11 persons. In other words, the main drawback in interpretation of the obtained results could be an incapacity to reveal significant differences due to low statistical power, even thought they may be present. This is the reason why it is difficult to confidently reveal the absence of significant changes in parameters like NT, SOD, catalase and CP (power ≤0.5). Therefore, we have also considered non-significant differences between the parameters. However, in case of TAC changes during rGH treatment (0.82), and in comparing TAC, NT and SOD between control and experimental groups, the power of the tests was relatively high for a correct interpretation of the obtained results.

We also believe that some discrepancy in the age of patients between control and experimental groups is not crucial for an interpretation of our results because participants of both groups were prepubertal children with similar physiology and similar parameters of the antioxidant system. It is common practice in studies dedicated to the antioxidant status to include prepubertal children aged 2–11 years in a single group (see e.g., *Erden-Inal, Sunal & Kanbak, 2002*; *Singh & Barjatiya, 2002*; *Llorente-Cantarero et al., 2012*).

We have demonstrated that parameters of the antioxidant system before treatment are not balanced in GHD children: TAC and NT levels are lower, while SOD activity is elevated albeit statistically non-significantly. The decreased TAC and NT level, the elevated SOD activity and the insignificantly raised level of ceruloplasmin registered in GHD children before rGH treatment, all indicate the presence of OS.

Obviously, a statistically significant resolution of only one parameter (TAC) during treatment indicates an insufficient prevention of the already ongoing OS in GHD children. However, a failure to reveal statically significant differences in TAC and NT (which characterize the total amount of antioxidants in the plasma and blood, respectively) after treatment between the groups, and also a non-significant decline in CP level (increase in CP level indicates inflammatory processes) still may suggest an improvement of the antioxidation state in children after therapy.

In our work we observed some evidence of blood antioxidant system improvements after rGH treatment, which is supported by the absence of a significant difference in TAC

and NT after treatment between the control and treatment groups (using one-way ANOVA with post-hoc Tukey HSD test, $p < 0.05$). Moreover, antioxidaton parameters such as TAC, NT and CP (but not SOD and catalase) came back to normal levels during treatment. Although these changes are small (not reaching statistical significance), we can conclude that these findings in general agree with data reported in *Mohn et al. (2005)*, who showed an improvement in the antioxidant status, evidenced by a decline in free radicals in GHD children after rGH treatment.

### Funding
This work was supported by Alfa-Endo Program of Charities Aid Foundation Russia, funded by Alfa-Group. The funders had no role in study design, data collection and analysis, decision to publish, or preparation of the manuscript.

### Grant Disclosures
The following grant information was disclosed by the authors:
Alfa-Group.

### Competing Interests
The authors declare there are no competing interests.

### Author Contributions
- Maria S. Pankratova conceived and designed the experiments, performed the experiments, analyzed the data, contributed reagents/materials/analysis tools, wrote the paper, prepared figures and/or tables, reviewed drafts of the paper.
- Adil A. Baizhumanov conceived and designed the experiments, performed the experiments, analyzed the data, contributed reagents/materials/analysis tools, reviewed drafts of the paper.
- Alexander I. Yusipovich conceived and designed the experiments, analyzed the data, wrote the paper, prepared figures and/or tables, reviewed drafts of the paper.
- Maria Faassen performed the experiments, analyzed the data, contributed reagents/materials/analysis tools, wrote the paper, reviewed drafts of the paper.
- Tatyana Yu. Shiryaeva performed the experiments, analyzed the data, contributed reagents/materials/analysis tools.
- Valentina A. Peterkova conceived and designed the experiments.
- Svetlana S. Kovalenko and Tatiana A. Kazakova performed the experiments.
- Georgy V. Maksimov conceived and designed the experiments, wrote the paper.

### Human Ethics
The following information was supplied relating to ethical approvals (i.e., approving body and any reference numbers):

The study was approved by the Ethics Committee of the Endocrinology Research Centre, Moscow, Russia Federation (reference number: 14). Written informed consent was obtained from the patients and/or their parents or legal guardians.

## Supplemental Information

Supplemental information for this article can be found online at http://dx.doi.org/10.7717/peerj.1055#supplemental-information.

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
