# Peer review of "Imbalance in the blood antioxidant system in growth hormone-deficient children before and after 1 year of recombinant growth hormone therapy"

_PeerJ, doi:10.7717/peerj.1055_

## Round 0.1 · original submission · Major Revisions

General comments:

The ms is difficult to read, because of extremely poor English style and grammar.

There is quite a large difference in ages of the GHD groups and controls. The authors do not provide information whether this may cause (part of) the difference between markers of the antioxydant system between GHD children and controls.

The changes after 12 months GH treatment were small, and only reached statistical significance for TAC. The direction of TAC and non-protein thiols was into the direction of controls, but this was not the case for SOD.

I have the impression that the number of patients is insufficient to answer the research question.

Specific comments:

Fig 1 is difficult to interpret, and offers the same information as the table. It can be omitted.
Table 1: It would be better if IGF-I would be expressed as SDS; the same applies to IGFBP-3

·

Basic reporting

The manuscript PeerJ-3977 “Imbalance in the blood antioxidant system in growth hormone-deficient children before and after 1 year of recombinant growth hormone therapy” asseses several parameters of the blood anti-oxidant system in control and GH-deficient (GHD) children (the last, before and at the end of a 12-month-period of GH therapy). They detect a baseline imbalance in GHD patients, as previously demonstrated in other studies, with significantly decreased anti-oxidant parameters and increased oxidant activity. At the end of the 1st year of GH therapy, total anti-oxidant capacity has significantly increased and is no longer different from that in controls while the SOD activity has not lowered. The conclusion would be that the anti-oxidant system improved while not sufficiently to completely normalise. Results go in the same direction than previously reported studies. The study is therefore not novel but is certainly interesting. Unfortunately, the manuscript is hampered by English language problems that even render difficult the understanding of some sentences. In that sense, the manuscript could be corrected.
A part from the English language the following observations would need improvement:
- 1) Abstract: reword Results description!
- 2) GHD diagnosis should be explained in more detail: following the 2 GH secretion stimulation tests and the image studies, GHD was diagnosed when GH peak values (in both tests?) were under 10 ng/ml (not pg/L). Was the type of GHD considered idiopathic, was it isolated in all patients?
- 3) IGF-1 assay type may be either an ELISA or a RIA but not both.
- 4) GH assay is not mentioned. It should be with the standard used for calibration. Depending on the assay, the limit of 10 ng/ml may be too high, and the majority of present-day assays are using cut-off values of 7.5-7.0 ng/ml to ascertain a GH secretion deficiency.
- 4) Results description is confusing, mainly the 2nd paragraph of the “Antioxidant status”: try to describe Table 2 results straightforward!
- 5) Discussion: the paragraph “After treatment .... “ is impossible to understand. Try to write it simply and clearly!
- 6) “The tendency to improve ...... during rGH therapy is evident, but ......”
- 7) Figure 1 legend: “.... were normalised on values of GHD patients ....”
- 8) Table 1 title is erroneous.
- 9) Mention which standards were used to transform anthropometric parameters into SDS.
- 10) Can the authors transform the IGF-1 and IGFBP-3 values into SDS for age and sex?

Experimental design

2) GHD diagnosis should be explained in more detail: following the 2 GH secretion stimulation tests and the image studies, GHD was diagnosed when GH peak values (in both tests?) were under 10 ng/ml (not pg/L). Was the type of GHD considered idiopathic, was it isolated in all patients?
- 3) IGF-1 assay type may be either an ELISA or a RIA but not both.
- 4) GH assay is not mentioned. It should be with the standard used for calibration. Depending on the assay, the limit of 10 ng/ml may be too high, and the majority of present-day assays are using cut-off values of 7.5-7.0 ng/ml to ascertain a GH secretion deficiency.
- 4) Results description is confusing, mainly the 2nd paragraph of the “Antioxidant status”: try to describe Table 2 results straightforward!

Validity of the findings

Findings are not novel

Additional comments

The manuscript PeerJ-3977 “Imbalance in the blood antioxidant system in growth hormone-deficient children before and after 1 year of recombinant growth hormone therapy” asseses several parameters of the blood anti-oxidant system in control and GH-deficient (GHD) children (the last, before and at the end of a 12-month-period of GH therapy). They detect a baseline imbalance in GHD patients, as previously demonstrated in other studies, with significantly decreased anti-oxidant parameters and increased oxidant activity. At the end of the 1st year of GH therapy, total anti-oxidant capacity has significantly increased and is no longer different from that in controls while the SOD activity has not lowered. The conclusion would be that the anti-oxidant system improved while not sufficiently to completely normalise. Results go in the same direction than previously reported studies. The study is therefore not novel but is certainly interesting. Unfortunately, the manuscript is hampered by English language problems that even render difficult the understanding of some sentences. In that sense, the manuscript could be corrected.
A part from the English language the following observations would need improvement:
- 1) Abstract: reword Results description!
- 2) GHD diagnosis should be explained in more detail: following the 2 GH secretion stimulation tests and the image studies, GHD was diagnosed when GH peak values (in both tests?) were under 10 ng/ml (not pg/L). Was the type of GHD considered idiopathic, was it isolated in all patients?
- 3) IGF-1 assay type may be either an ELISA or a RIA but not both.
- 4) GH assay is not mentioned. It should be with the standard used for calibration. Depending on the assay, the limit of 10 ng/ml may be too high, and the majority of present-day assays are using cut-off values of 7.5-7.0 ng/ml to ascertain a GH secretion deficiency.
- 4) Results description is confusing, mainly the 2nd paragraph of the “Antioxidant status”: try to describe Table 2 results straightforward!
- 5) Discussion: the paragraph “After treatment .... “ is impossible to understand. Try to write it simply and clearly!
- 6) “The tendency to improve ...... during rGH therapy is evident, but ......”
- 7) Figure 1 legend: “.... were normalised on values of GHD patients ....”
- 8) Table 1 title is erroneous.
- 9) Mention which standards were used to transform anthropometric parameters into SDS.
- 10) Can the authors transform the IGF-1 and IGFBP-3 values into SDS for age and sex?

---

## Round 0.2 · Minor Revisions

Please see editorial revisions of text and tables (attached as a PDF and also available to you as the original Word docs).

---

## Round 0.3 · accepted · Accept

Thanks for the further improvements.